# Directional phyllotactic bias in calatheas (*Goeppertia*, Marantaceae): A citizen science approach

Benjamin Durrington[1], Fiona Chong[2,3] (iD), Daniel H. Chitwood[4,5] (iD),

Twitter Calathea Poll Participants and iNaturalist Citizen Scientists

[1]Department of Integrative Biology, University of Texas at Austin, Austin, Texas , USA; [2]Department of Biological & Marine Sciences, University of Hull, Hull, United Kingdom; [3]Energy and Environment Institute, University of Hull, Hull, United Kingdom; [4]Department of Horticulture, Michigan State University, East Lansing, Michigan, USA; [5]Department of Computational Mathematics, Science & Engineering, Michigan State University, East Lansing, Michigan, USA

## Citizen Science

**Keywords:**
calathea; *Goeppertia*; handedness; leaf asymmetry; phyllotaxy; spirality.

Benjamin Durrington and Fiona Chong contributed equally to the manuscript.

**Author for correspondence:** D. Chitwood, E-mail: dhchitwood@gmail.com

## Abstract

Lateral organs arranged in spiral phyllotaxy are separated by the golden angle, ≈137.5°, leading to chirality: either clockwise or counter-clockwise. In some species, leaves are asymmetric such that they are smaller and curved towards the side ascending the phyllotactic spiral. As such, these asymmetries lead to mirroring of leaf shapes in plants of opposite phyllotactic handedness. Previous reports had suggested that the pin-stripe calathea (*Goeppertia ornata*) may be exclusively of one phyllotactic direction, counter-clockwise, but had limited sampling to a single population. Here, we use a citizen science approach leveraging a social media poll, internet image searches, in-person verification at nurseries in four countries and digitally-curated, research-grade observations to demonstrate that calatheas (*Goeppertia spp.*) around the world are biased towards counter-clockwise phyllotaxy. The possibility that this bias is genetic and its implications for models of phyllotaxy that assume handedness is stochastically specified in equal proportions is discussed.

## 1. Introduction

Phyllotaxy is the arrangement of leaves and other lateral organs on a plant. The most common phyllotactic pattern is spiral, in which lateral organs are separated by the golden angle, ≈137.5°, derived from the Fibonacci sequence (Jean, 2009; Prusinkiewicz & Lindenmayer, 2012). We ascend the spiral of a plant starting at the base of the shoot axis where the oldest leaves are located and follow successively younger leaves up towards the shoot apex where the youngest leaves are initiating. Here, we refer to the phyllotactic direction (the chirality) of the ascending spiral as either clockwise or counter-clockwise (Figure 1). We use the term *chirality* for phyllotactic direction (either clockwise or counter-clockwise) to distinguish it from the *asymmetry* of leaves (left- or right-handed), which we refer to later, to avoid confusing distinct but interconnected phenomena. Another reason we use clockwise and counter-clockwise to describe phyllotactic chirality is because we are defining phyllotactic direction by *ascending* the phyllotactic spiral (whereas left- and right-handed helical conventions describe *descending* along spirals, see Edwards et al., 2007). Early studies across numerous, diverse plant species not only found that the ratio of clockwise and counter-clockwise individuals in a population tend to be 50:50 (De Vries, 1903; Koriba, 1914; Ikeno, 1923), but also that phyllotactic direction is not heritable (Imai, 1927). Although not spiral phyllotaxy, a special case of seeds from alternating rows of maize ears and other grass inflorescences producing seedlings in which the left or right side of the first leaf would overlap the other was reported by Compton (1912). In this case, too, phyllotactic direction is not heritable and the result of developmental circumstance (i.e., which alternating row the seed is found on the ear determining leaf overlap in the seedling). Another potential influence on phyllotactic direction is the environment. Considering phyllotactic direction in a number of herbaceous and woody species in the South Atlantic region of the United States, Allard (1951) found that overall results tended towards a 50:50 ratio in aggregate. However,

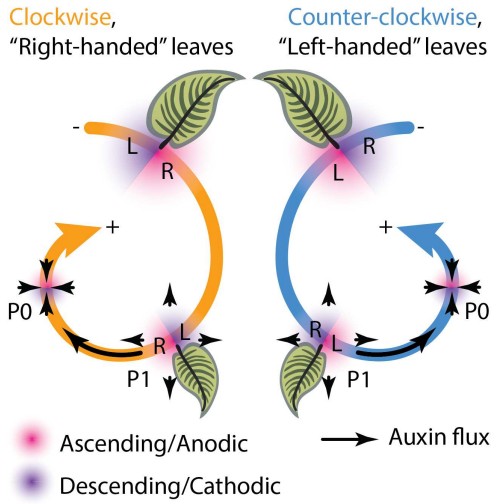

**Clockwise,** "Right-handed" leaves

**Counter-clockwise,** "Left-handed" leaves

Ascending/Anodic
Descending/Cathodic
Auxin flux

**Fig. 1.** Phyllotactic leaf asymmetry. Phyllotactic chirality, either clockwise (orange) or counter-clockwise (blue), is determined ascending the phyllotactic spiral from the base of the plant to its tip. In clockwise phyllotaxy, the ascending (anodic) side (magenta) of the leaf is right and the descending (cathodic) side (purple) of the leaf is left, while in counter-clockwise the relationship is reversed. Because the incipient leaf (P0) is an auxin sink while older primordia (such as P1) are auxin sources, the descending side of the leaf is exposed to higher auxin levels, leading to leaf asymmetries that manifest on opposite sides of leaves in clockwise and counter-clockwise phyllotaxy. Note that leaf asymmetry, which is either left- or right-handed, is distinct from phyllotactic chirality (clockwise or counter-clockwise). Leaf asymmetry is a consequence of phyllotactic chirality, but it is a distinct morphological phenomenon. Hence, we refer to right-handed leaves arising from clockwise phyllotaxy and vice versa.

in keeping track of the same species sampled in different locations or times a number of sub-samples significantly deviated from a 50:50 ratio, at odds with the overall results. Even when performed correctly, the interpretation of statistical results can yield false positives or represent under-sampled environmental effects. Especially when untangling genetic and environmental effects, it is imperative that sampling is broad and robust to avoid spurious conclusions.

Spiral phyllotactic handedness impacts asymmetry at an organismal level through the same mechanism by which phyllotaxy itself arises: the plant hormone auxin (Kuhlemeier, 2007). Other mechanisms besides auxin, notably microtubules and cell wall composition, affect chirality at an organismal level as well. In the case of microtubules, the handedness of cortical microtubules guides the directionality of cellulose microfibrils, leading to right-handed cell twisting that propagates to the organismal level (Furutani et al., 2000; Thitamadee et al., 2002; Buschmann et al., 2004); whereas in a microtubule-independent manner, rhamnose-containing cell wall polymers suppress left-handed helical twisting in petals and roots (Saffer et al., 2017; Saffer & Irish, 2018). Unlike the aforementioned mechanisms, only auxin links leaf asymmetry with phyllotactic direction. Auxin is sufficient to direct the location of leaf primordium initiation (Reinhardt et al., 2000). When considered with the intracellular localization of the auxin efflux carrier PIN-FORMED1 (PIN1) within the shoot apical meristem (Reinhardt et al., 2003) that directs auxin towards neighbouring cells with high auxin concentrations (Benková et al., 2003), computational modelling can recapitulate spiral phyllotactic patterning (Smith et al., 2006). Similar mechanisms of self-organization directed by auxin and PIN1 operate after leaf initiation, patterning venation (Scarpella et al., 2006), the margin (Bilsborough et al., 2011) and leaflets (Koenig et al., 2009). A fundamental relationship between

leaf primordia and auxin efflux leading to spiral patterning is that the incipient leaf (P0, plastochron 0) acts as an auxin sink and that older primordia (such as the neighbouring next oldest leaf primordium, P1) are auxin sources. Because of the handedness of spiral phyllotaxy, the ascending (anodic) side of the P0 (closer to the shoot apical meristem) might receive less auxin than the descending (cathodic) side (away from the shoot apical meristem and closer to the P1) because of the proximity of the P1, an auxin source (Figure 1). Considering the effects of auxin on leaf venation and margin development, the above scenario might create morphological asymmetries in the mature leaf. Because the ascending side of the leaf is the right-hand side in clockwise phyllotaxis and the left-hand side in counter-clockwise phyllotaxis, such asymmetry would be expected to manifest as mirror images in plants with opposite phyllotactic directions (Figure 1). Morphometric analysis of tomato and *Arabidopsis* leaves, auxin localization in leaf primordia and their curvature and computational modelling of auxin asymmetries in the shoot apical meristem all confirm auxin-dependent mirroring of leaf asymmetry arising from opposite phyllotactic directions (Chitwood et al., 2012a).

Independent of the above auxin-based mechanism and subtle statistical morphometric leaf asymmetries, Korn (2006) described easily observable morphological features that corresponded with the ascending (anodic) side of leaves, and thus phyllotactic direction. In *Syngonium podophyllum*, the midrib curves and leaf coils towards the ascending side of the leaf; in *Acalypha virginica*, the axillary bud on the ascending side is smaller than the descending side; in *Croton variegatus* 'Banana', a secondary blade is only initiated on the ascending side of leaves; and in *Aglaonema crispum*, the smaller half of the leaf is ascending (Korn, 2006). As the ascending side is the right side of the leaf in clockwise phyllotaxis and the left side in counter-clockwise phyllotaxis, phyllotactic direction was also recorded. All of the above species had plants with shoot apices of both phyllotactic directions (i.e., both clockwise and counter-clockwise phyllotactic chirality were present among plants). Leaf asymmetry was also described for *Goeppertia ornata* (referred to as *Calathea ornata* in Korn, 2006), in which the ascending side of leaves was smaller than the descending side. Remarkably, of the 26 shoots examined arising from nine plants, all were counter-clockwise (Korn, 2006). Assuming a null hypothesis of a 50:50 ratio, the $p$-value of all 26 shoots displaying counter-clockwise phyllotaxy is $3.414 \times 10^{-7}$ and 0.0027 for all nine plants (Chi-squared goodness of fit test). If all shoots are of the same phyllotactic handedness, then any leaf asymmetries will be exclusively restricted to the same physical side. Korn (2006) reports all 132 leaves from the 26 shoots examined having smaller ascending sides, which together with the knowledge that all shoots were counter-clockwise means that it was exclusively the left sides of the *G. ornata* (*C. ornata*, Korn, 2006) leaves that were smaller than the right side. This report of exclusive counter-clockwise handedness in a species—with strong effects on leaf asymmetry—has important implications for phyllotactic theory, in which the null hypothesis assumes that spiral phyllotaxy is stochastically patterned in a 50:50 ratio.

Here, we leverage a citizen science approach to see if indeed calatheas (*Goeppertia spp.*) are biased towards a counter-clockwise phyllotactic direction. Using the popularity of calatheas as houseplants and social media, we initiated a Twitter poll to which 105 people responded. The results of the poll showed a strong bias of calathea plants exhibiting counter-clockwise compared to clockwise phyllotaxy, as inferred from the handedness of leaf shape. To verify the results beyond the poll, we undertook a number of approaches. First, we examined plants from Google image searches

**Table 1.** A list of our study taxa in its old classification under the genus *Calathea*, and their new and correct binomial nomenclature after Borchsenius et al. (2012)

| Previously known as | Most recent binomial nomenclature *sensu* Borchsenius et al. (2012) |
|---|---|
| *Calathea concinna* (W.Bull) K.Schum. | *Goeppertia concinna* (W.Bull) Borchs. & S.Suárez |
| *Calathea lancifolia* (Boom) | *Goeppertia lancifolia* (Boom) Borchs. & S.Suárez |
| *Calathea louisae* (Gagnep.) | *Goeppertia louisae* (Gagnep.) Borchs. & S.Suárez |
| *Calathea majestica* (Linden) H.Kenn. | *Goeppertia majestica* (Linden) Borchs. & S.Suárez |
| *Calathea makoyana* (É.Morren) | *Goeppertia makoyana* (É.Morren) Borchs. & S.Suárez |
| *Calathea orbifolia* (Linden) H.Kenn. | *Goeppertia orbifolia* (Linden) Borchs. & S.Suárez |
| *Calathea ornata* (Linden) Körn. | *Goeppertia ornata* (Linden) Borchs. & S.Suárez |
| *Calathea picturata* K.Koch & Linden | *Goeppertia picturata* (K.Koch & Linden) Borchs. & S.Suárez |
| *Calathea roseopicta* (Linden ex Lem.) Regel | *Goeppertia roseopicta* (Linden ex Lem.) Borchs. & S.Suárez |
| *Calathea rufibarba* Fenzl | *Goeppertia rufibarba* (Fenzl) Borchs. & S.Suárez |
| *Calathea warscewiczii* (L.Mathieu ex Planch.) Planch. & Linden | *Goeppertia warscewiczii* (L.Mathieu ex Planch.) Borchs. & Suárez |
| *Calathea zebrina* (Sims) Lindl. | *Goeppertia zebrina* Nees |

of calathea species and scored phyllotactic direction based on the appearance of leaves. Next, we verified phyllotactic direction in nurseries across four different countries and two continents. Finally, using research-grade photos collected from citizen scientists from iNaturalist obtained through the Global Biodiversity Information Facility (GBIF) we further verify that the bias in phyllotactic direction extends across the world and to calatheas growing in natural populations. Together, our results demonstrate that cultivated calatheas (and likely natural populations, too) exhibit a strong bias in phyllotactic direction, across numerous independent observations across the world and upon personal verification by the authors. Hypotheses regarding the origin of the bias are discussed, with implications for a possible genetic origin of phyllotactic handedness in calatheas.

## 2. Materials and methods

While referring to our study taxa, we aim to reflect the most recent taxonomic changes and promote ease of use. *Calathea,* as formerly defined, is polyphyletic, and was recently split into two main groups (Borchsenius et al., 2012). One clade (*Calathea II*) is more closely related to *Ischnosiphon* than the rest of *Calathea*, and retains the generic name. A second clade (*Calathea I*), including all of our study species, is now included in the genus *Goeppertia*. We still refer to these generally as 'calatheas' here and in the Twitter poll in order to facilitate name recognition but adopt the most recent binomials throughout (Table 1).

A Twitter poll was initiated July 20, 2020. An explanatory figure was posted with the poll that asked readers 'Is your Calathea a Lefty or a Righty?', showing example calathea leaves of each hand (Figure 2a). The leaf images were taken from Korn (2006), with attribution stating that the figure had been modified by reflection for demonstration (the left-handed images were the original image; the right-handed leaves were modified by reflection). Left and right-handed designations were selected to make it easier for the public to determine the handedness of their leaves and do not correspond to helical conventions of handedness (and in fact, they can be interpreted as opposite of helical conventions of left and right used for the twining of vines; e.g., see Edwards et al., 2007). Using left- and right-handed to describe leaf asymmetry also separates these terms from phyllotactic chirality, for which

we reserve clockwise and counter-clockwise, for clarity (Figure 1). Hence, we refer to right-handed leaves arising from clockwise phyllotaxy and vice versa. Left-handed leaves lend themselves to such a label and were described as 'Left side smaller' and 'Leaf curved towards left' with corresponding arrows pointing left. The poll itself reiterated the description of right- and left-handed leaves (Figure 2b). Participants in the poll could respond 'My plant is left-handed', 'My plant is right-handed', or 'Neither of the above'. The poll lasted for 1 week, after which it was closed, and results were made publicly available.

To verify the Twitter poll results, calathea images from Google image search were used by the authors (Figure 3). Five calathea species (that are popular as houseplants) were queried: *G. lancifolia*, *G. makoyana*, *G. orbifolia*, *G. ornata* and *G. roseopicta*. The first 50 non-redundant images from the search were saved and the links from where they originated were recorded. Images of *Aloe polyphylla* were similarly curated as a control. The three authors scored each image as −1, 0, +1 for left-handed, neither or right-handed, respectively. 'Neither' is a category that encompasses plants with both left- and right-handed leaves in equal ratios, leaves that do not have handedness, or plants where left- or right-handed determination could not be made for any number of reasons. The three scores were averaged. *A. polyphylla* images were scored once and not averaged as phyllotactic handedness was obvious.

In-person verification of plants was undertaken by the authors by examining calathea plants at local nurseries. All available calathea plants at each location were examined and a photo was taken with a marker to ensure that manipulation by reflection had not taken place. Each author scored the respective plants they examined as −1, 0, +1 for left-handed, neither and right-handed, respectively. Plants were examined in nurseries at the following locations: La Tinaja, Guanajuato Mexico (Vivero de La Tinaja; 20.489814, −101.212225); Hull, UK (Plant & Paint; 53.73944985, −0.333975936); Brussels, Belgium (Dille & Kamille, Brussel Grasmarkt; 50.84788842, 4.35271669); Austin, Texas, USA (Tillery Street Plant Co.; 30.26114649, −97.70419208) (Figures 4 and 5, Table 1). The number of plants and species at each location was as follows: Mexico, 28 plants total, 8 *G. makoyana*, 14 *G. ornata*, 5 *G. picturata*, 1 *G. zebrina*; United Kingdom, 106 plants total, 6 *G. concinna*, 15 *G. lancifolia*, 4 *G. louisae*, 31 *G. majestica*, 6 *G. orbifolia*, 1 *G. ornata*, 7 *G. picturata*, 23 *G. roseopicta*, 2 *G. rufibarba*, 1 *G.*

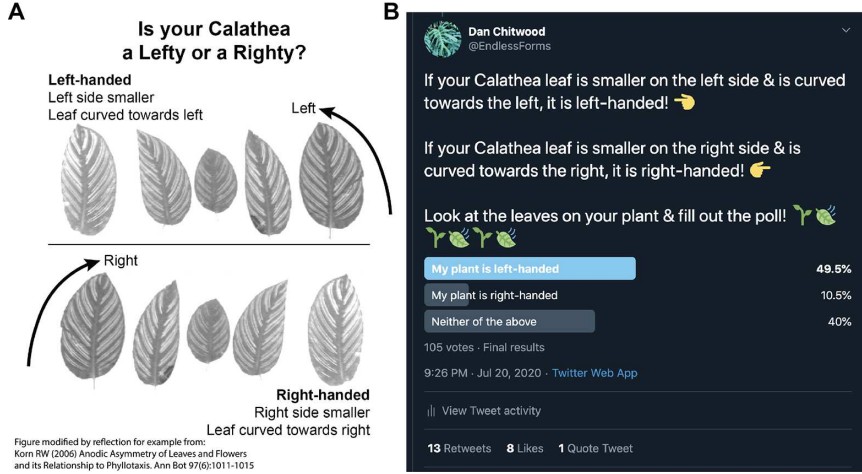

**Fig. 2.** Twitter poll and results. (a) Diagram showing poll participants example calathea leaves to orient them to leaf handedness. This figure was modified by reflection from the original figure of Korn (2006) for demonstration (left-handed leaf images are the original images; right-handed leaf images have been reflected). A note of the modification from Korn (2006) was made in the original image (bottom left hand corner). (b) Final poll results, made public after the poll ran for a week, are shown.

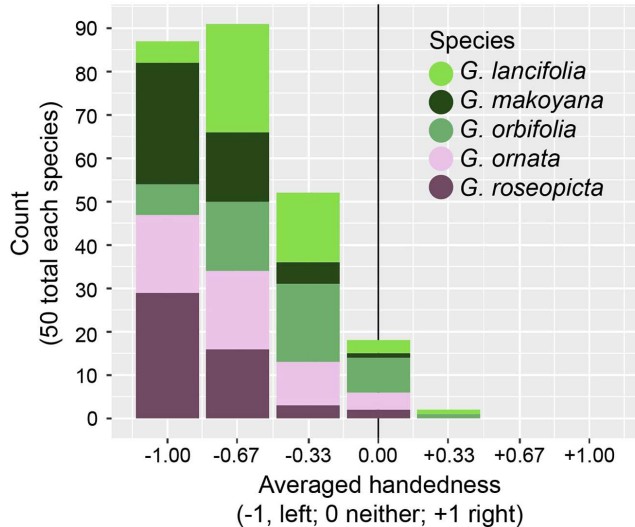

**Fig. 3.** Google image search results. The first 50 images for each species (*G. lancifolia, G. makoyana, G. orbifolia, G. ornata* and *G. roseopicta*) by Google image search were scored for handedness as left (−1), neither (0) or right (+1) by each of the three authors and averaged. Shown are the counts for each species for each averaged score. Colors indicating each species are provided in legend.

*warscewiczii*, 10 *G. zebrina*; Belgium, 8 plants total, 4 *G. concinna*, 1 *G. ornata*, 2 *G. roseopicta*, 1 *G. rufibarba*; United States, 17 plants total, 16 *G. louisae* 'Maui Queen', 1 *G. zebrina*.

Digitally-curated, research-grade photos taken from around the world by citizen scientists through iNaturalist were obtained through the GBIF. iNaturalist photos available through GBIF are 'research-grade', meaning there is a photo, date and coordinates for the occurrence and that the community agrees on an identification (Ueda, 2020). Although cultivated calatheas are present, unlike the Twitter poll, Google image searches and nursery inspections the GBIF data allow access to calatheas grown in the wild. To verify the range distribution of wild-growing calatheas, all instances of 'Goeppertia Nees' (including iNaturalist photos, herbarium specimens and other occurrences) were downloaded from GBIF (GBIF.org, December 30, 2020). 3,768 occurrences with coordinates (excluding data points from null island and in the Arctic

ocean) were plotted (Figure 5) showing the vast majority of samples falling within the native range of *Goeppertia* species in Central and South America. Fifty-six iNaturalist photos of calathea plants in natural environments downloaded through GBIF were scored for phyllotactic chirality (Figure 4). The number of photos of each species is as follows: 22 *G. zebrina*, 22 *G. ornata*, 4 *G. warscewiczii*, 2 *G. majestica*, 2 *G. makoyana*, 1 *G. roseopicta*, 1 *G. louisae*, 1 *G. picturata*, 1 *G. rufibarba*. The number of photos from each country is as follows: 13 Costa Rica, 11 Peru, 7 Ecuador, 4 Panama, 3 Brazil, 2 Colombia, 2 Trinidad and Tobago, 2 Mexico, 2 Malaysia, 2 Singapore, 1 Guyana, 1 Nicaragua, 1 Puerto Rico, 1 Hong Kong, 1 India, 1 Indonesia, 1 Réunion, 1 Seychelles. In total, 45 out of 56 photos (80.4%) are from Central and South America (excluding Mexico) or the Caribbean.

Statistical analyses were performed in R (R Core Team, 2019). All visualizations were made using ggplot2 (Wickham, 2016). For tests of bias in handedness from Korn (2006), the Twitter poll, nursery inspections and iNaturalist observations, Chi-squared goodness of fit test was used with the chisq.test() function. For the Google image search results, a two-tailed Wilcoxon signed rank test was used with the wilcox.test() function.

## 3. Results

The Twitter poll yielded 105 responses: 49.5%, 52 responses 'My plant is left-handed'; 10.5%, 11 responses 'My plant is right-handed'; 40%, 42 responses 'Neither of the above' (Figure 2b). Disregarding 'Neither of the above' responses and testing against a null hypothesis of 50:50 left-to-right handed, Chi-squared goodness of fit test rejects the null hypothesis ($p = 2.398 \times 10^{-7}$).

For the scoring of handedness from Google image search results, five species with 50 images each were scored and averaged between the three authors (Figure 3). A two-sided Wilcoxon signed rank test was used to test the null hypothesis that the average of the distribution for each species was 0 (neither). For each species, the null hypothesis was rejected with the following *p*-values: *G. lancifolia*, $p = 1.808 \times 10^{-9}$; *G. makoyana*, $p = 3.885 \times 10^{-10}$; *G. orbifolia*, $p = 1.931 \times 10^{-8}$; *G. ornata*, $p = 2.041 \times 10^{-9}$; *G. roseopicta*, $p = 4.879 \times 10^{-10}$. The distributions for each species were heavily skewed towards −1, indicating left-handed leaf asymmetry and counter-clockwise phyllotactic chirality. *A. polyphylla* was used

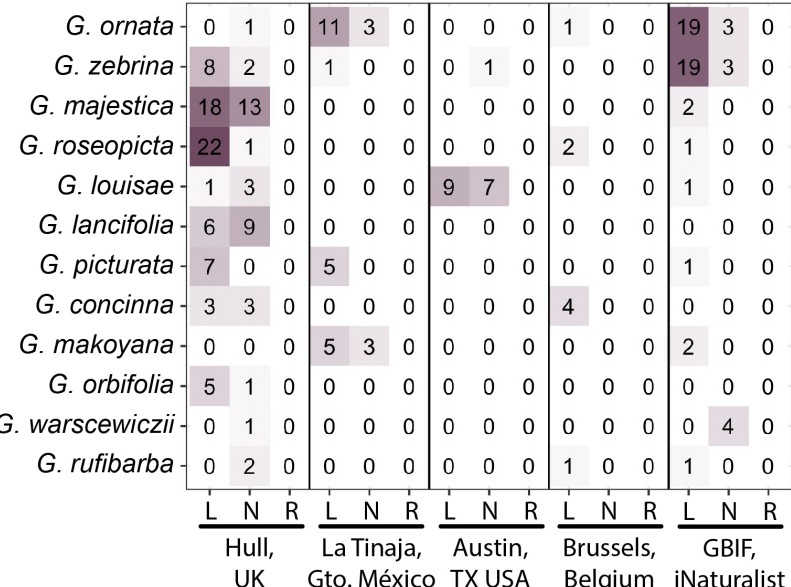

**Fig. 4.** Phyllotactic chirality in nursery-inspected plants and iNaturalist observations. Numbers of *Goeppertia* species (vertical axis) with left- (L), neither (N) or right-handedness (R) at four different nursery locations and citizen scientist iNaturalist observations across the world downloaded through GBIF (horizontal axis). Color intensity (purple) indicates the number of plants (indicated by text).

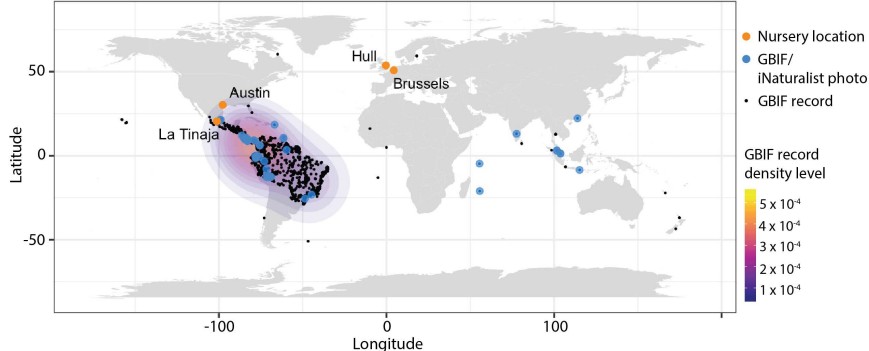

**Fig. 5.** Locations of observations used in this study. A world map showing the locations of the four nurseries calathea plants were inspected by the authors (orange points) and the 56 locations of iNaturalist observations made by citizen scientists (blue points). The locations of 3,768 GBIF occurrences with coordinates are plotted as small black points with a corresponding density heat map to indicate where *Goeppertia* Nees occurrences are most prevalent in the native range (Central and South American and the Caribbean).

as a control to ensure that images downloaded from the internet had not been reflected. The phyllotactic direction in this species is obvious and images were qualitatively scored as clockwise (28 images) and counter-clockwise (22 images). A Chi-squared goodness of fit test does not reject the null hypothesis of a 50:50 ratio ($p$ = .3961).

Handedness of plants was verified in person at nurseries in four countries (Figures 4 and 5; Table 1). Because the authors could verify plant handedness in person, each plant was simply scored as left-handed (−1), neither (0), or right-handed (+1). A Chi-squared test was used with a null hypothesis assuming equal proportions of each category. In all locations the null hypothesis was rejected: La Tinaja, Gto. Mexico, $n$ = 28 and $p$ = 9.592 × 10$^{-7}$; Hull, UK, $n$ = 106 and $p$ = 8.689 × 10$^{-16}$; Brussels, Belgium, $n$ = 8 and $p$ = .0003355; Austin, Texas, USA, $n$ = 17 and $p$ = .01365. If plants at all locations are combined, 159 plants were measured with a $p$-value of 4.034227 × 10$^{-25}$. Overall, 109 plants were scored as left-handed, 50 as neither and 0 as right-handed.

Research-grade citizen scientist photos from iNaturalist (Ueda, 2020) of calathea plants from around the world were downloaded

through the GBIF (GBIF.org, December 30, 2020) (Figure 5). Photos were scored the same as the nursery inspections: left-handed (−1), neither (0) or right-handed (+1) (Figure 4). A Chi-squared test was used with a null hypothesis assuming equal proportions of each category. Of the 56 photos, 46 were scored as left-handed, 10 as neither and 0 as right-handed. The null hypothesis was rejected: $n$ = 56, $p$ = 2.408605 × 10$^{-14}$.

## 4. Discussion

The Twitter poll provided a global platform to sample truly independent responses (Figure 2). Respondents did not know the results of the poll before they responded, and once results were publicized, the poll closed. There was no way respondents could be biased by previous responses. Unlike a common garden experiment in which local environmental effects are highly controlled for but the extrapolation of results to different environments is limited, the Twitter poll results are globally sampled. Each respondent purchased or propagated and grew their calathea plant in a truly unique environment. For an easy to measure, qualitative trait,

such a poll allows for a robust survey of the overall proportion of handedness in calatheas from numerous, independent sources across the world. Nonetheless, there are caveats. Although strong leaf asymmetry and clear instructions make it easy to infer phyllotactic direction, participants may have made mistakes, or at worst fabricated results. In the case of fabrication, the question is such that responses would be random (there is no reason to select one response over another). Twitter users that wanted to see results without participating may have voted for either 'Neither of the above' or 'My plant is left-handed'/'My plant is right-handed' with equal probability. Compared to results from the Google image search, in-person inspection of plants at nurseries by the authors, and a survey of iNaturalist photos from citizen scientists, the Twitter poll has an exceptionally high proportion of responses for 'neither' (40%) as well as the ratio of right-to-left handed (17.5:82.5), indicating that more than likely responses to these categories are inflated. By discounting the 'neither' responses altogether and focusing on testing the ratio of right-to-left against a null hypothesis of 50:50, the null hypothesis is rejected ($p = 2.398 \times 10^{-7}$). This result is likely robust, as the above-mentioned sources of error act to support the null hypothesis, not the alternative.

Three methods were used to verify the Twitter poll result. The first was Google image searches of specific calathea species (Figure 3). This method still relies on others taking photos of plants and assumes that the photos have not been reflected. Like the Twitter poll, it should be noted that the source of error (the photographer reflecting the photo) works in support of the null hypothesis, not the alternative. Additionally, the *A. polyphylla* results indicate that, at least for this species, reflection did not affect the observed ratio of 50:50 in phyllotactic handedness. Unlike the Twitter poll, the authors themselves scored each of the images, averaging across their responses (−1 for left, 0 for neither and +1 for right). The null hypothesis that the mean of the distribution was zero was rejected for each species, and there was a strong left-handed bias for each species. The second method to verify results was in-person visits by each of the authors to nurseries across four countries and two continents (Figures 4 and 5, Table 1). Here, the authors could rely on their own expertise to score the plants. Among 159 plants, none were scored as right-handed and all were scored as either left-handed or neither. The third method to verify results was by examining research-grade photos from citizen scientists through iNaturalist and GBIF (Ueda, 2020; GBIF.org, December 30, 2020). The most important advantages of this method compared to the others are accessible coordinate data and inclusion of both wild and cultivated calatheas. The vast majority of GBIF occurrences for *Goeppertia* Nees fall within the native range of calatheas, and approximately 80% of the 56 examined photos fall within the native range as well (Figure 5). Among the 56 photo observations, none were scored as right-handed and all were scored as either left-handed or neither, similar to the nursery inspection results (Figure 4).

Together the original results of Korn (2006), the Twitter poll, scoring of images from Google image search, and in person inspection of nursery plants, we conclude that leaves in cultivated calatheas throughout the world are highly skewed to left-handed, indicating counter-clockwise phyllotaxy. Considering the research-grade photos from iNaturalist citizen scientists, we also believe that it is highly probable calathea plants in the wild are highly skewed to left-handed, indicating counter-clockwise phyllotaxy, but this statement requires further investigation. We only reject (strongly) the null hypothesis that the ratio of clockwise to counter-clockwise

is 50:50 in calatheas, as with the present data we cannot support the hypothesis that phyllotaxy in calatheas is exclusively counter-clockwise.

Besides the original report by Korn (2006) of exclusive phyllotactic direction in *G. ornata* (*C. ornata*), another report is in banana (*Musa*; Skutch, 1927; 1930) where it is reported to be clockwise: 'The spiral in which the leaves are arranged is always left-handed; it rises from the right to the left of an observer facing the stem' (Skutch, 1927). As an effect of the fixed phyllotactic direction, leaf morphology is impacted as well: 'Every banana leaf I have ever examined closely was rolled in the same way, the right half covered by the left' (Skutch, 1930). These statements need to be verified, similar to those for calatheas here, and there is at least one published work with illustrations of leaf asymmetry of both hands contradicting this claim (Argent, 1976). Interestingly it is indicated that exclusive phyllotactic direction is diagnostic of different banana species. If true, it is suspicious that *Goeppertia* and *Musa* are both members of the Zingiberales. One hypothesis is that because of rhizomatous growth and propagation by division, biased phyllotactic handedness is preserved through propagation of limited wild collections of these taxa. However, most commercial calathea and banana propagation is now through tissue culture (Banerjee & de Langhe, 1985; Chen & McConnell, 2006). Considering the production of shoot apical meristems *de novo* during tissue culture, it is expected that phyllotactic direction would be reset. Further, seed propagation in calatheas is possible and does occur. Finally, as described by a major distributor of calatheas in Europe, Gebr. Valstar, horticultural collections of wild *Calathea* are regularly made during expeditions to Brazil, providing new material for production and breeding new cultivars (https://www.thursd.com/growers/gebr-valstar/, accessed December 31, 2020). It is unlikely that so many species and varieties would exhibit strong phyllotactic biases in the same direction as observed here simply because of chance, limited collection from wild populations, or the way they are propagated. A genetic mechanism specifying the bias in phyllotactic direction is more likely.

A genetic bias in handedness is not unprecedented. Twining in vines is predominantly counter-clockwise (right-handed helices; Edwards et al., 2007). Similar to *Goeppertia*, this bias is observed throughout the world, indicating a strong genetic component independent of environment. Another example of directional asymmetry is *Alstroemeria*, in which the resupinate leaves exclusively twist in a counter-clockwise direction. The phyllotactic ratio is unaffected and 50:50 in *Alstroemeria*, but leaves from clockwise phyllotactic plants are smaller than those from counter-clockwise plants, perhaps due to conflict arising between counter-clockwise resupination and clockwise phyllotaxy (Chitwood et al., 2012b). There is also a molecular basis for directional asymmetries, where microtubule mutants and mutations affecting the composition of the cell wall in *Arabidopsis* create directional helices, but phyllotactic direction is not impacted (Furutani et al., 2000; Hashimoto, 2002; Thitamadee et al., 2002; Buschmann et al., 2004; Saffer et al., 2017; Saffer & Irish, 2018). This is despite a connection between microtubules and phyllotaxy (Heisler et al., 2010). If genetic, the fixed phyllotactic handedness in *Goeppertia* (and potentially other members of Zingiberales) is singularly unique. Across disparate species, phyllotaxy is reported to occur equally in opposite directions (De Vries, 1903; Koriba, 1914; Ikeno, 1923; Imai, 1927), a non-heritable trait that is stochastically determined. The widespread bias in independently grown calathea plants across the world reported here from citizen scientists, which verify the original observations by Korn (2006), demonstrate unknown

mechanisms at play, and that current models of phyllotaxy are incomplete.

## Acknowledgements

FC thanks Lara Roberts from Plant & Paint (Hull, UK) for supplying calathea plants for photographing. The authors thank reviewers for insightful and constructive comments that improved the quality of the manuscript.

**Financial support.** This project was supported by the USDA National Institute of Food and Agriculture, and by Michigan State University AgBioResearch.

**Conflict of interest.** The authors declare no conflicts of interest.

**Authorship contributions.** DHC conceived the study. BD, FC and DHC designed the study. BD, FC and DHC conducted data gathering and phenotyping. Twitter Calathea Poll Participants and iNaturalist citizen scientists provided data. DHC performed statistical analyses. BD, FC and DHC wrote, edited and reviewed the manuscript.

**Data availability statement.** All data, code and photos (from Google image searches, in-person nursery inspections by the authors and iNaturalist/GBIF citizen science photos) necessary to reproduce this work can be found on figshare at https://doi.org/10.6084/m9.figshare.13140434.v1 (containing original data) and https://doi.org/10.6084/m9.figshare.13513176.v1 (containing revised results).

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
