## [Reviewer Report]

Dear Editors of Quantitative Plant Biology,

Please find submitted the manuscript “Directional phyllotactic bias in calatheas (Goeppertia, Marantaceae): a citizen science approach” by Benjamin Durrington, Fiona Chong, Dan Chitwood, and the Twitter Calathea Poll Participants as a Citizen Science Article to Quantitative Plant Biology.

Spiral phyllotaxy is assumed to occur in roughly a 50:50 clockwise to counterclockwise ratio across plants. This assumption has been tested numerous times in old literature across species and locations and phyllotactic direction has been shown not to be heritable. Korn (2006) found that when leaves are asymmetric, the side to which they curve and are smaller on tends to be on the ascending (anodic) side of the genetic spiral (such that leaves are smaller on the right side in clockwise phyllotaxy and vice versa). In most species, the side of the leaf on which the asymmetry presents itself is 50:50, following the direction of the phyllotactic handedness of the plant. One exception Korn noted was Goeppertia ornata (previously Calathea ornata), in which the 26 shoots for the 9 plants he examined were all counterclockwise in direction. If calathea plants were exclusively or biased towards counterclockwise phyllotaxy, it would indicate a potential example in which phyllotactic direction is genetically specified.

Here, we verify the results of Korn using a citizen science approach. A twitter poll with over 105 independent responses from around the world, verification of phyllotactic direction using leaf asymmetry in 250 photographs from five Goeppertia species from a google image search, and in-person verification of phyllotactic direction by the authors in 13 species from four countries and two continents in which not a single clockwise phyllotactic plant among 159 specimens was detected, all point to a strong phyllotactic bias across calatheas that deviates from a 50:50 ratio. Because a role of environment cannot be discounted, the broad, global sampling enabled using social media and internet-based citizen approaches produces a robust result that demonstrates a genetic basis for phyllotactic handedness. As current phyllotactic models assume that spiral phyllotactic direction is not genetically but stochastically specified, our results point to a previously unconsidered phenomenon that has implications for the understanding of phyllotaxy in general. We believe for this reason this manuscript is uniquely qualified for the Citizen Science article type in Quantitative Plant Biology, as the robustness of the result is derived from hundreds of independent samples from around the world, reported by citizen scientists.

All data, code, and images used in this manuscript can be found at figshare at https://figshare.com/articles/dataset/Directional_phyllotactic_bias_in_Calathea_spp_Marantaceae_a_citizen_science_approach/13140434

We would like to recommend the following reviewers:

Joyce Chery (Cornell, jgc235@cornell.edu), because of her expertise in twining and plant asymmetry

Chelsea Specht (Cornell, cds266@cornell.edu), because of her expertise in Zingiberales phyllotaxy

Siobhan Braybrook (UCLA, siobhanb@ucla.edu), because of her expertise in phyllotaxy-derived leaf asymmetry

Thank you for your consideration, on behalf of the authors,

Dan Chitwood

References:

Korn, R. W. (2006). Anodic asymmetry of leaves and flowers and its relationship to phyllotaxis. Annals of Botany, 97(6), 1011-1015.

---

## [Reviewer Report]

*Comments to Author*: L65- Is “genetic” necessary?

L76- developmental constraint on what?

L89- In really enjoyed the discussion about auxin, and I think the introduction would be strengthened by adding in key references from the robust literature of handedness from Arabidopsis which has identified that the chirality of the cortical microtubules, guide the directionality of the cellulose microfibrils, directing cell expansion, and ultimately chirality. Mutants such as spr1-3, spr1-6, spr2-2, tua4, tua5 highlight the significance of MTs, while rhm1-3, rhm1-2 highlight the role of the cell wall composition in determining chirality.

Figure1. I think its logical to separate the chirality of phyllotaxy from the handedness of the leaves, as these are separate issues, however this terminology is confusing because chirality is usually poses are either left or right handed. To mitigate the potential of confusion, please be explicit in the text and in the caption that you are describing two separate processes. Perhaps using the same terminology would help, for example: left-handed phytollotaxy + right handed leaves

L130- Do you mean each of those species are variable (have both L and R chirality)?

L170- Deeply appreciate the taxonomic history!

L175- Ah yes, this would be a good place to be super clear that you used r and l handed leaves as the proxy for phyllotaxy, which is actually the opposite. As its written now, its sorrrrt of clear, but could be improved.

Figure 3. I think this would be better represented as a histogram, as -1 to 1 is distribution rather than discrete categories.

L208: Be more explicit about the locality of these nurseries. ‘

L267: Do you have access to the locality data of the twitter poll? Are calathea mostly vegetatively propagated or from seed? If the former, do you know if there is a few large distributers of the horticultural crop, in which most people have purchased from? This would mean that folks are more likely than not to have plants from the same genetic stock. A stretch? Maybe… but possible! If all of this is untrue, and all plants are propagated from see, it would be good to mention this to eliminate this minor doubt.

L326- oh la la- so they are vegetatively propagated. Why would you expect this to reset the phyllotaxy?

L330- provide a citation for this.

---

## [Reviewer Report]

*Comments to Author*: In the present study, the authors reported the investigation of the directional phyllotactic bias in calatheas by using a citizen science approach, and found that leaves in calatheas are highly skewed to left-handed, indicating the calatheas are predominately biased towards counter-clockwise phyllotaxy. Overall, this is an interesting study, the scientific approaches and results proceed in a logical manner and are generally convincing. The findings may help to uncover the genetic mechanism underlying the directional phyllotactic bias and handedness. However, there are several issues that need to be addressed as listed below.

1. The sample size in the Twitter poll is a little bit small. I would suggest the authors to expand the poll scope by using additional social media to collect more responses covering diverse countries and regions as much as possible.

2. It seems that 10.5% votes in the Twitter poll is “my plant is right-handed”. I am wondering whether the corresponding participators (or all participators) provided the evidences to prove their votes.

3. This manuscript reported that cultivated calatheas exhibit a strong bias in phyllotactic direction. How about the wild or landrace varieties of calatheas? I would suggest the authors to verify the directional phyllotactic bias in the wild and landrace varieties of calatheas if possible.

---

## [Reviewer Report]

*Comments to Author*: One reviewer raised questions regarding the Twitter poll, including the poll size and evidence supporting participants' selections.

---

## [Reviewer Report]

Dear Quantitative Plant Biology,

Thank you for considering and reviewing our manuscript. We found the reviewers' comments most helpful. To address concerns of the Twitter poll size, locality data of responses, the question of phyllotactic bias in wild calatheas, and photographic evidence, we turned to iNaturalist data, which addresses these points. The inclusion of wild calathea data (for which we observe likely the same phyllotactic bias found in cultivated calatheas) strengthens the hypothesis that that the phyllotactic bias we observe is genetically specified. 

Thank you again for considering our manuscript.

Dr. Dan Chitwood